# Estimation method for impact force source in thick coal seams and its application in rockburst risk quantification

Quande Wei[1], Quanjie Zhu[1]*, Guangyu Yang[2,3], Dongsheng Jiang[4], Yi Liu[5], Yingnan Hao[4], Qilin Hao[6]

1 School of Emergency Technology and Management, North China Institute of Science and Technology, Sanhe, Hebei, China, 2 China Coal Mining Research Institute Co. Ltd., Beijing, China, 3 Coal Mining & Designing Department, Tiandi Science & Technology Co.Ltd., Beijing, China, 4 School of Mine Safety, North China Institute of Science and Technology, Sanhe, Hebei, China, 5 Information Institute of Ministry of Emergency Management, Beijing, China, 6 Yima Coal Industry Group Corporation Ltd., Yima, Henan, China

☯ These authors contributed equally to this work and are considered co-first authors.
* zhqj2016@ncist.edu.cn

## Abstract

Accurately calculating the stress state and its distribution is key to preventing coal burst. Taking a typical mine with extremely thick coal seams and coal bursts as an example, methods such as stress estimation, numerical computation, theoretical analysis, and on-site measurements were used to study the estimation method for coal burst force sources and the quantitative evaluation of coal burst risk under extremely thick coal seams beneath massive gravel rock. The main research content includes: (1) Establishing a coal burst force source estimation model for extremely thick coal seams through the estimation of transmitted stress from massive gravel rock, self-weight stress of the overlying strata, tectonic stress quantitative analysis, and mining disturbance stress estimation; (2) Proposing a calculation method for the fracture angle and strike span parameters required for stress estimation based on microseismic monitoring technology, and comparing and verifying the effectiveness of rapid estimation methods for strike and dip support pressures; (3) Based on the above methods, proposing a quantitative evaluation method for coal burst risk based on stress estimation. The practical results demonstrate that the proposed method enables rapid analysis of impact force sources, accurate assessment of coal burst risk levels, and establishes a foundation for timely and precise early warning of coal burst disasters.

## 1. Introduction

Rockbursts (also known as coal rockbursts) are one of the most destructive dynamic hazards in deep coal mining, posing a severe threat to mine safety and production

**Data availability statement:** All relevant data are within the manuscript and its Supporting Information files.

**Funding:** This work is financially supported by the Key Science and Technology Program of the Ministry of Emergency Management (2024EMST070702), the Natural Science Foundation of Hebei Province of China (E2023508021, E2024508004) and the Fundamental Research Funds for the Central University (3142024005, 3142021002). The funders had no role in study design, data collection and analysis, decision to publish, or preparation of the manuscript.

**Competing interests:** The authors have declared that no competing interests exist.

efficiency. The primary factors affecting rockbursts include the physical and mechanical properties of coal seams, the mechanical characteristics of the overlying rock layers, mining methods, excavation disturbances, stress distribution, and geological structure. Among these, stress factors play a key role. Rockbursts occur when the coal rock body becomes suddenly unstable, and their essence is the violent release of elastic strain energy [1–3]. As the core manifestation of mining-induced stress redistribution, support pressure directly determines the formation and triggering mechanisms of rockbursts. Therefore, rockbursts are fundamentally a "stress" issue [3]. The distribution characteristics of support pressure in mining areas are crucial for mine support design and rockburst prevention. There is a correlation between the movement of roof rock layers, the evolution of support pressure, and roof disasters. The movement of the roof rock layer is the root cause of support pressure evolution and roof disasters, and the evolution of support pressure is a precursor to roof disasters. The movement and evolution of the overlying rock layers are the fundamental reasons for the development of underground dynamic disasters. Understanding the movement laws of overlying rock layers and their evolutionary disaster mechanisms is of significant importance for controlling dynamic disasters [4].

In recent years, research on rockbursts has mainly focused on the mechanical properties of coal seams, excavation disturbances, and stress distribution analysis [2,5,6]. It is clear that accurately calculating the stress state and its distribution is the key to preventing rockbursts [7,8]. According to different research methods, the study of support pressure in mining areas can be divided into several stages, including theoretical research, numerical simulations [3,9], laboratory experiments, and field measurements [10–12]. In the theoretical research phase, Qian Minggao [13,14], from the perspectives of elastoplastic theory and damage mechanics, proposed the key layer theory, which unified the study of rock layer movement from the coal seam to the key layer and provided a theoretical basis for studying the formation and instability of mining-induced overlying rock structures. This theory has helped achieve disaster prevention and control through understanding rock layer movement laws. Jiang Fuxing et al.[15–17], using similar material simulation, numerical simulation, and field monitoring methods, studied the distribution patterns, influence range, peak positions of support pressure, and their changes with roof movement. Zhang Haifeng et al.[18] used geophysical techniques and real-time monitoring systems (microseismic, stress, etc.) to reveal the support pressure distribution characteristics in mining areas, successfully applying them to rockburst disaster prevention. Ju Jinfeng et al.[19] used surface drilling in-situ monitoring to study the movement laws of overlying rock layers under deep well and large mining height conditions. These studies provide practical technical means for revealing the issues related to mining-induced rock movement.

In engineering practice, the evolution of support pressure can be analyzed and monitored to evaluate the likelihood of roof disasters and provide early warnings. The distribution characteristics of support pressure depend on the range and movement characteristics of the roof rock layer. The research progress on these characteristics is related to the form of roof disasters, and the range of the roof rock layers involved changes as the forms of roof disasters change. Therefore, the author attempts to

quantitatively calculate the distribution of support pressure caused by the mining of a retreat mining face and assess the risk of rockburst occurrence. The engineering value of quantitative support pressure calculation mainly lies in three aspects: 1) accurate risk assessment, 2) optimization of preventive measures, and 3) upgrading the early warning system. The quantitative estimation of the rockburst source can provide the theoretical basis for these three aspects, which not only holds theoretical research value but also has important practical significance for the safe production of mines [12,14].

Based on the above analysis, this study introduces the stress superposition theory, combined with microseismic monitoring and numerical simulations, to propose a rockburst mechanism analysis and early warning method for ultra-thick coal seams under ultra-thick conglomerate, filling a research gap and offering strong novelty and practicality. Specifically, this study uses elastoplastic mechanics, rock mechanics, and overlying rock spatial structure theories, combined with microseismic monitoring and UDEC numerical simulation technology, to explore the rockburst force estimation methods for ultra-thick coal seams under ultra-thick conglomerates. The main innovations of this research are: first, through the superposition stress estimation method, an in-depth study of the mechanical interaction between the ultra-thick conglomerate layer and the coal seam was carried out, and a rapid estimation model for support pressure was established; second, a quantitative evaluation method for rockburst risk based on stress superposition was proposed. These studies not only provide new ideas for preventing rockbursts but also provide a scientific basis for safety assessment and risk early warning in coal mining.

## 2. Project overview and background introduction

The 21221 working face is located in the western wing of the deep excavation area of the 21st downhill mining zone at Qianqiu Coal Mine in Yima, Henan. The design location of this working face and the surrounding mining conditions are shown in Fig 1. Meanwhile, the currently recovering working faces include the 21141 working face in the western wing of the mining area and the 21172 working face in the eastern wing. The area between the 21141 working face and the 21181

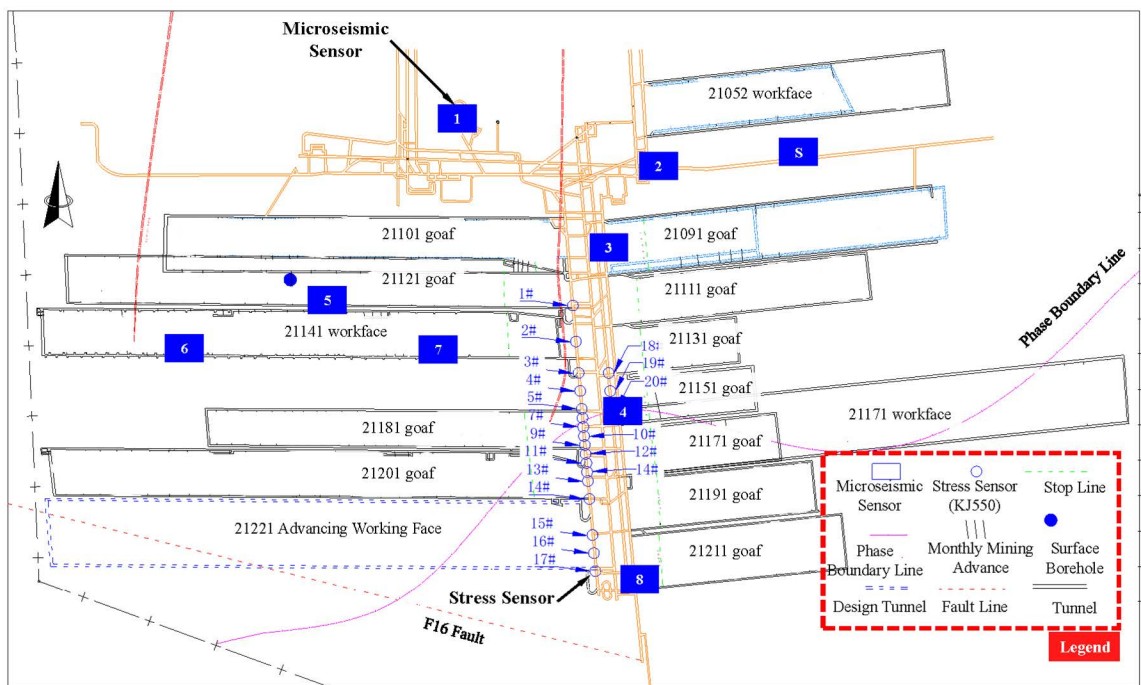

**Fig 1. Location of the 21221 Working Face and Layout of Monitoring System Measurement Points.**

goaf has yet to be mined. The average mining depth of the 21221 working face is 800 m, and the coal pillar width between the 21201 goaf and the 21221 working face upper drift is 5 m. The lower drift side is the solid coal yet to be mined in the deep section of the mining area. The design strike length of the working face is 1500 m, and the dip length is 180 m. Both the upper and lower drifts advance along the floor but leave a bottom coal thickness of 0.3 m to 2 m.

The 21221 working face is located in the merged area of the 2–1 coal seam and the 2–3 coal seam, referred to as the 'Second Coal.' After impact tendency testing by the Beijing Coal Science Institute, it was determined that both the Second Coal and its roof strata exhibit weak coal rockburst tendency.

The total coal thickness ranges from 3.89 m to 11.10 m, with a coal seam dip angle between 3° and 13°. The roof and floor strata of the coal seam, along with their thickness. The floor strata beneath the coal seam consist of alternating layers of carbonaceous mudstone, mudstone, sandstone, and fine sandstone. The roof strata above the coal seam consist of mudstone, gravel, sandstone, fine sandstone, and mixed mudstone layers, with a massive thick gravel layer on top. The surface loess layer is relatively thin, and in some areas, the gravel layer has been exposed. The main geological structures within the 21221 working face include the phase boundary and the F16 reverse fault. The F16 regional reverse fault has a shallow dip angle of 75° and a deep dip angle ranging from 15° to 35°, with a drop of 50 m to 500 m.

With the increase in mining depth, the geological conditions in the mining area become more complex, particularly in the mining of extremely thick coal seams under thick gravel rock cover. The occurrence mechanisms, mechanical properties, and prevention/control methods of coal rockburst have become key research topics. The coal rockburst mechanism under the combined action of thick gravel rock and extremely thick coal seams not only involves the mechanical properties of the coal seam itself but also includes the mechanical response of the overlying strata, as well as the complexity of stress transfer and accumulation. Therefore, understanding the characteristics of thick gravel rock and extremely thick coal seams, combined with the interaction between the coal seam and the overlying strata, is of great theoretical and practical significance for effectively preventing and controlling coal rockburst.

## 3. Estimation of coal rockburst force sources under extremely thick coal seams and massive gravel rock

The magnitude of the total stress resulting from the superimposed self-weight stress of the overlying strata, stress transferred from the massive gravel rock, tectonic stress, and mining disturbance stress is a key factor in the occurrence of coal rockburst in extremely thick coal seams under massive gravel rock. Therefore, this section starts by examining the stress generated by various force source influence factors, and studies the estimation method for stress transferred from the massive gravel rock and the quantitative analysis method for tectonic stress. A multi-factor superimposed stress estimation method is used to couple and estimate the stress generated by each influencing factor, exploring the distribution pattern of the total superimposed stress.

The principle of using high-precision microseismic technology to obtain the fracture angle and strike span lies in the fact that when rock layers are subjected to stress, they fracture or experience minor fractures, which are accompanied by the generation of microseismic signals. By deploying a microseismic sensor array, we are able to capture these vibration signals in real-time and, through the time, frequency, and spatial distribution of the signals, locate the position of the fracture and the extent of crack propagation. By analyzing the propagation path of the seismic signals and the energy distribution, we can estimate the fracture angle (break angle) and strike span of the rock layers. These parameters provide foundational data for the stress estimation model of coal burst in this study. From the fracture angle perspective, a larger angle $a$ leads to more pronounced stress transfer into deeper coal zones. Regarding the strike span, parameter $L$ governs the energy release magnitude during key stratum rupture, with $L > 100$ m significantly increasing the probability of high-energy seismic events. Against this backdrop, through the spatial positioning capabilities of microseismic monitoring technology, we are able to accurately determine important parameters such as the fracture angle ($a$) and strike span ($L$) of the rock layers. The stress estimation workflow is illustrated in the flowchart below (Fig 2).

**Fig 2. Flowchart of stress estimation workflow.**

## 3.1. Estimation method for stress transferred from massive gravel rock

The Qianqiu Coal Mine in Henan is a typical example of a coal mine with extremely thick coal seams under massive gravel rock. Based on surface subsidence observation data from Qianqiu Coal Mine, there has been no significant surface subsidence in the mining area under the extremely thick coal seams, indicating that the massive gravel rock layer has not yet fractured and exhibits a large suspended roof. The mine is still in the non-fully mined stage. At the same time, microseismic monitoring data from the 21141 working face of Qianqiu Coal Mine also reflects that the overlying massive gravel rock layer has not fractured.

Through a comprehensive analysis of the mining area's strata structure, mining conditions, and on-site microseismic data, a spatial structure of the overlying strata close to the actual conditions can be obtained. A static support pressure estimation model for the dip and strike directions under non-fully mined conditions is established, as shown in Fig 3. In the support pressure estimation model, the overlying strata of the mining area are estimated by using key strata as the estimation units, while the self-weight of non-key strata is applied as evenly distributed load on the key strata. The key strata are divided into $n$ groups, from the coal seam to the surface. Under non-fully mined conditions, some of the key strata undergo delamination as mining progresses, and the line connecting the front end of delamination of each key layer is approximately regarded as the strata fracture line. The angle between this fracture line and the horizontal line is called the strata fracture angle $\alpha$.

When establishing the static support pressure estimation model for strike/dip under insufficient mining conditions, it is necessary to clarify the key assumptions and their theoretical basis. Specifically, they are as follows: (1) In terms of load distribution, two assumptions are introduced: the dominant key layer assumption and the uniform load simplification. The

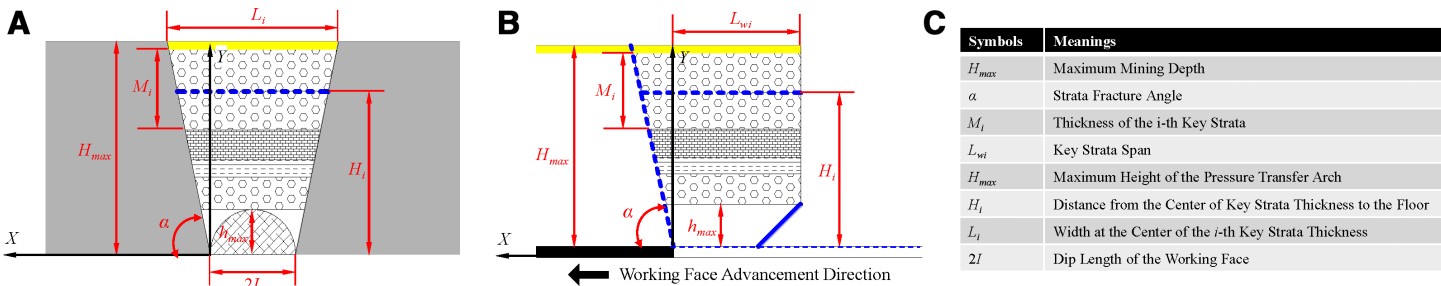

**Fig 3. Support Pressure Estimation Models for Extremely Thick Coal Seams under Massive Gravel Rock: (a) Lateral Support Pressure Estimation Model; (b) Strike Support Pressure Estimation Model; (c) Explanation of Symbols in the Figure.**

former assumes that only the key layer bears the overlying rock structure load, while the self-weight of the non-key layer is converted into a uniform load ($q = \gamma h$). This assumption is based on the key layer theory, which confirms that thick hard rock layers control over 90% of the load transfer. The latter is based on St. Venant's principle, which mainly considers that the load of the non-key layer acts uniformly on the underlying key layer. (2) In terms of boundary condition assumptions, assumptions such as linearization of the fracture line and static stress field are introduced. The former assumes that the line connecting the front end of the layer separation of each key layer is simplified to a straight line (with a constant fracture angle α), which is consistent with both field conditions and results from similar simulation experiments. (3) In terms of layer behavior, it is assumed that ideal delamination behavior and fracture angle consistency exist. Considering practical engineering applications, this study, based on the above analysis and key assumptions, conducts research on the quantitative estimation model for impact force sources.

### 3.1.1. Estimation and calculation of lateral support pressure.

The lateral support pressure estimation model is shown in Fig 3(a). The overlying strata structure mainly consists of high-position strata and low-position strata. The high-position strata are composed of the unfractured massive gravel rock layer and the surface loess layer. These layers have a very long subsidence period and minimal subsidence, and can be regarded as an evenly distributed load continuously acting on the underlying strata and coal body. The low-position strata structure consists of strata groups that undergo periodic fracturing as mining progresses. In the low-position strata structure, the dip cantilever length of each key strata group is relatively independent of the mining advance speed.

The combined effect of high-position and low-position strata forms the transmitted stress of the overlying strata. The formula derivation process for calculating lateral support pressure is as follows:

① Stress Increment Superposition Estimation Process.

Under non-fully mined conditions, assume that the weight transmitted by the $i$-th key strata to one side of the working face is half of its weight. The stress increment transmitted to the coal seam is approximately distributed in an isosceles triangle. The stress increment transmitted by the $i$-th key strata to one side of the working face is given by equation (1):

$$\Delta \sigma i = \begin{cases} \sigma_{\max i} \tan \alpha\, x / H_i & x \in [0, H_i \cot \alpha] \\ 2\sigma_{\max i}(1 - x/2H_i \tan \alpha) & x \in [H_i \cot \alpha,\ 2H_i \cot \alpha] \\ 0 & x \in (2Hi \cot \alpha,\ \infty) \end{cases}, \tag{1}$$

Where: σmax $i$—The maximum support pressure generated by the $i$-th key strata on the coal seam, MPa; $H_i$— Distance from the center of the $i$-th key strata to the coal seam floor, m; 2l—Width of the goaf, m; $M_i$ — Thickness of the $i$-th key strata, m; α—Strata fracture angle, °; $Q_i$— Half the weight of the exposed portion of the $i$-th key strata in the goaf, N/m; γ—The rock's unit weight, N/m³; $L_i$— Cantilever length of the $i$-th key strata's center in the goaf, m. $\sigma_{\max i}$ can be expressed by equation (2):

$$\sigma_{\max i} = M_i \gamma \left( 1 + l \left/ \left( \left( l + \frac{M_i}{2} + \sum_1^{i-1} M_j \right) \tan \alpha \right) \right. \right), \tag{2}$$

After superimposing the stress increments from all $n$ key layers, the total stress increment Δσ can be obtained.

② Estimation of Support Pressure from Self-Weight **σ$_q$**.

The estimation of support pressure from self-weight $\sigma_q$ is given by equation (3):

$$\sigma_q = \begin{cases} \gamma l & x \in [0, l \cot\alpha] \\ \gamma x \tan \alpha & x \in [l \cot\alpha, H_{\max} \cot\alpha] \\ \gamma H_{\max} & x \in [H_{\max} \cot\alpha, \infty) \end{cases}, \tag{3}$$

Microseismic monitoring, numerical simulations, and theoretical derivations can all provide the strata fracture angle, which can be optimized based on specific conditions in practical applications. When the calculated strata fracture angle deviates significantly from conventional mining theory, a comparison and verification using all three methods should be conducted to increase the reliability of the lateral support pressure estimation.

**3.1.2. Estimation model for strike support pressure.** Through theoretical research and on-site measurement analysis, a strike support pressure estimation model is established, as shown in Fig 3(b). This model is similar to the dip support pressure estimation model, but microseismic data from the site shows that the cantilever length of the key strata groups in the low-position strata structure is positively correlated with the mining advance speed within a certain range.

The calculation steps for strike support pressure are the same as those for lateral support pressure estimation. However, the following parameter estimation formula differs:

$$H_i = h_{max} + M_i/2 + \sum_{1}^{i-1} M_j,$$

(4)

Where: $H_i$— Distance from the $i$-th key strata group to the coal seam, m; $h_{max}$ — Maximum height of the working face's pressure transmission arch, m; $M_i$— Thickness of the $i$-th strata group, m; σ— Strata fracture angle, °; $Q_i$—Approximate weight of half of the exposed portion of the $i$-th key strata in the goaf, N/m; γ— The rock's unit weight, N/m³; $L_i$— Cantilever length of the $i$-th key strata's center in the goaf, m.

The maximum support pressure generated by the $i$-th key strata group on the coal seam in front of the working face is given by equation (5):

$$\sigma_{max\,i} = \frac{(L_{wi}\tan\alpha + H_i)M_i\gamma}{(2h_{max} + M_i + 2\sum_{1}^{i-1} M_j)} = \frac{(L_{wi}\tan\alpha + H_i)M_i\gamma}{2(h_{max} + M_i/2 + \sum_{1}^{i-1} M_j)} = (\frac{L_{wi}\tan\alpha}{H_i} + 1)\frac{M_i\gamma}{2},$$

(5)

The self-weight generated support pressure $\sigma_q$ is estimated as shown in equation (6):

$$\sigma_q = \begin{cases} \gamma h_{max} & x \in [0, h_{max}\cot\alpha] \\ \gamma\tan\alpha \times x & x \in [h_{max}\cot\alpha, H_{max}\cot\alpha] \\ \gamma H_{max} & x \in [H_{max}\cot\alpha, \infty] \end{cases},$$

(6)

## 3.2. Tectonic stress quantitative analysis

Tectonic stress is one of the force sources leading to coal rockburst in extremely thick coal seams under massive gravel rock. Tectonic stress mainly includes folding (synclines, anticlines), faults, phase boundary zones, etc [20]. Many domestic and international experts and scholars have conducted extensive research on the analysis of various tectonic stresses, achieving a large body of research results [21,22]. However, tectonic stress is difficult to calculate accurately, and most methods estimate the direction and magnitude of the force sources through macroscopic analysis. The results of these analyses often do not meet the needs of on-site engineering, thus requiring a more quantitative analysis of tectonic stress.

**3.2.1. Stress analysis of the F16 reverse fault and phase boundary zone.** Detailed parameters of the F16 reverse fault have not yet been measured, but according to geological reports and on-site engineering exposure at Qianqiu Coal Mine, the F16 reverse fault has disrupted the integrity of the Mianchi-Yima syncline, with a length of approximately 45 km. The strike is nearly east-west, dipping slightly to the south-east, with a shallow dip angle of 75° and a deep dip angle ranging from 15° to 35°. The drop is between 50 m and 450 m. Stress analysis of the F16 reverse fault has already been conducted [23]. Fig 4(a) shows the boundary fault section already exposed at Qianqiu Coal Mine. It can be determined

that the F16 reverse fault is a stress accumulation fault, and the residual tectonic stress causing horizontal movement in the fault still exists. The coal body on the hanging wall of the fault shows an upright phenomenon, so there is significant horizontal thrust on the coal body of the hanging wall. Under the combined effects of self-weight and horizontal forces, high shear stress exists along the fault plane. The coal body on the footwall is subject to significant sliding forces due to multiple working faces being mined out, and the deeper coal body also carries the self-weight stress from the overlying strata. Therefore, the deep coal body experiences combined effects from the horizontal thrust of the fault, shear forces from fault plane sliding, self-weight stress from overlying strata, and the sliding force from the goaf. High stress exists in both the vertical and horizontal directions.

**3.2.2. Quantitative analysis of tectonic stress increment.** There is no scientific, mature research that can be applied for the quantitative calculation of tectonic stress, such as from faults or phase boundary zones. To quantitatively analyze the distribution range and influence of tectonic stress, a method for quantitative analysis of tectonic stress is explored: a Cartesian coordinate system is established, and the distance $l$ from the influencing factors to the coordinate origin is provided based on the on-site geological conditions. Stress distribution laws are determined through similar model tests, numerical simulations, on-site measurements, analogies, and other methods, resulting in stress increment coefficients $k$ and influence range $L$. The stress increment coefficient $k$ and the influence range on each side of the fault plane/phase boundary line are different. However, for convenience in on-site engineering estimation, they are approximated to be the same. To determine the influence range and degree of tectonic stress on coal rockburst, first, the distance $l$ between the fault plane/phase boundary line and the analysis area is determined based on existing geological data. Then, through engineering experience, the stress increment peak value $k\sigma_z(x)$ and the influence range $L$ are estimated. Tectonic stress is approximated to follow a segmented linear distribution, as shown in Fig 4(b). From Fig 4(b), the stress distribution relationship on the fault/phase boundary side can be determined by equation (7).

$$\Delta\sigma_z(x) = \begin{cases} \frac{k\sigma_z(x)}{L/2}\left(x - \left(l - \frac{L}{2}\right)\right) & \left(l - \frac{L}{2}, l\right) \\ \left[1 + \frac{2}{L}(x - l)\right]k\sigma_Z(x) & \left(l, l + \frac{L}{2}\right) \end{cases}, \tag{7}$$

Based on the experience and numerical calculations of on-site engineering technicians from multiple extremely thick coal seam mines in Shandong, Henan, and Heilongjiang, the stress concentration coefficient and stress distribution range caused by the fault structure have been summarized and are provided in Table 1.

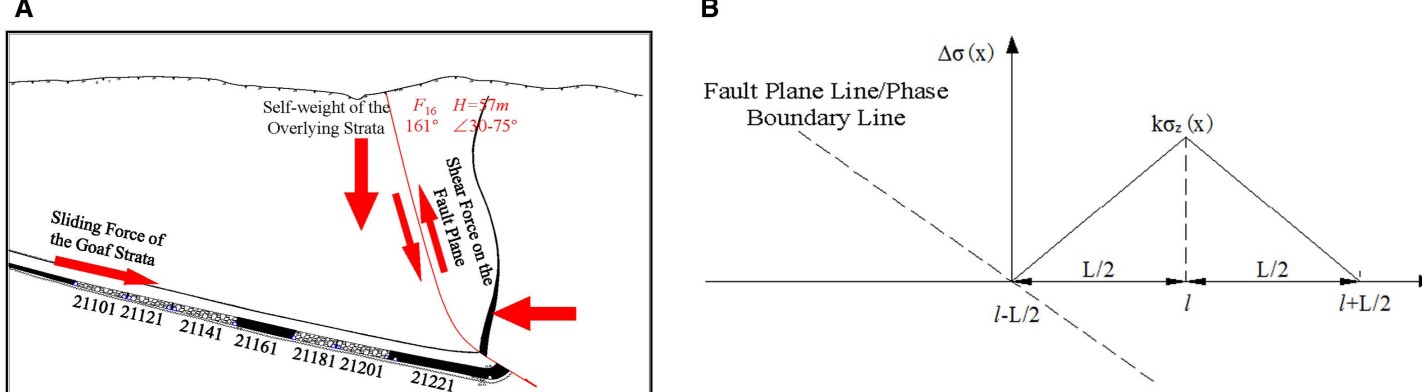

**Fig 4. Stress Analysis and Stress Increment Distribution of the F16 Reverse Fault Section: (a) Stress Analysis Diagram of the F16 Reverse Fault Section; (b) Stress Increment Distribution Relationship on the Fault/Phase Boundary Side.**

 

**Table 1. Empirical Values for Fault Drop, Stress Concentration Coefficient, and Influence Range.**

| Fault Drop (m) | Stress Increment Coefficient (k) | Double-Sided Influence Range 2L (m) | Single-Sided Influence Range L (m) |
| --- | --- | --- | --- |
| 0 ~ 5 | 1.2 | 28 | 14 |
| 5 ~ 10 | 1.3 | 36 | 18 |
| 10 ~ 30 | 1.4 | 50 | 25 |
| >30 | 1.5 | ≥120 | ≥60 |

### 3.3. Superimposed stress estimation method

The basic idea of the superimposed stress estimation method is to estimate the stress increment $\Delta\sigma_{zi}(x)$ and self-weight stress $\sigma_i(x)$ generated by each coal rockburst factor at a certain location. By superimposing these stress increments, the total stress $\sigma_{sum}(x)$ on the coal body is obtained. Essentially, there exists a stress mutation area near the influence factors that induce coal rockburst. The judgment of rockburst or coal rockburst in rock mechanics is commonly done using $\sigma/[\sigma_c]$, and the specific judgment criteria vary with changing conditions. Assuming the ratio of the total stress $\sigma_{sum}(x)$ to the uniaxial compressive strength $\sigma_c$ of the coal body is the dynamic stress ratio $d(x)$, the dynamic stress ratio is used as the standard for assessing coal rockburst risk. Based on empirical data, the risk zone and degree of risk are divided.

$$d(x) = \frac{\sigma_{sum}(x)}{[\sigma_c]} = \frac{\sum_{i=1}^{n} \Delta\sigma_{zi}(x) + \sigma_z(x)}{[\sigma_c]} = \frac{\sigma_z(x)\left(1 + \sum_{i=1}^{n} k_i(x)\right)}{[\sigma_c]},$$

(8)

Where: $d(x)$— Dynamic stress ratio, the ratio of the superimposed total stress to the uniaxial compressive strength of the coal rock; $\sigma_{sum}(x)$— Total stress at a point, MPa; $\sigma_z(x)$— Self-weight stress at a point, Mpa; $[\sigma_c]$— Uniaxial compressive strength, Mpa; $\Delta\sigma_{zi}(x)$— Stress increment generated by the $i$-th coal rockburst influencing factor, Mpa; $k_i(x)$ — Stress increment coefficient generated by the $i$-th factor.

## 4. Estimation of fracture angle and strike span based on microseismic monitoring

Obtaining the fracture angle of the strata, which is close to the actual field conditions, is a necessary condition for estimating the transmitted stress from massive gravel rock. However, due to the large thickness and high strength of the gravel rock layer, after coal seam mining, the massive gravel rock forms a large-scale suspended roof structure, and no significant surface subsidence occurs. It is difficult to estimate the fracture angle through surface subsidence observation data. Therefore, a method based on microseismic measurement data to estimate the fracture angle is proposed. Combining the support pressure estimation method and the spatial structure of the overlying strata, a rapid estimation method for the support pressure applicable to working faces with gently inclined, extremely thick coal seams under massive gravel rock is studied. This can also be referred to as a rapid estimation method for transmitted stress from massive gravel rock.

### 4.1. Fracture angle estimation method

Typically, the fracture angle of strata is estimated using surface subsidence data [24], but this method requires significant human resources, material resources, and time. Particularly under conditions of extremely thick coal seams under massive gravel rock, there are cases where no significant surface subsidence occurs after mining, making it difficult to estimate the fracture angle from subsidence data. High-precision microseismic monitoring technology is applied to continuously monitor and collect real-time fracture events in the overlying strata. By analyzing the three-dimensional spatial profile of microseismic events obtained from on-site measurements, the fracture and delamination positions (strata fracture lines) in the dip direction of the overlying strata can be determined. From these, the fracture angle of the strata can be estimated based on the horizontal and vertical distances of specific points along the fracture line.

Based on this background, a method for estimating the fracture angle of strata using high-precision microseismic monitoring technology is proposed [11]. The steps can be summarized as follows: (1) Select microseismic events within a specific time range, not less than 3 months, and group all events by energy level (grouped into four categories: $E \leq 10^4$ J, $10^4$ J $< E \leq 10^5$ J, $10^5$ J $< E \leq 10^6$ J and $\geq 10^6$ J. (2) Project the microseismic events onto the dip profile according to the groupings. Then, based on the comprehensive stratigraphic column, determine the altitude (depth coordinates) of key strata groups and estimate the average strata fracture line (strata movement line). (3) Based on the fracture line and the strata data from the comprehensive stratigraphic column, calculate the fracture angle for each group of microseismic events, and finally compute the average fracture angle of the strata as the fracture angle under the established mining conditions for the working face.

To illustrate the estimation process, the 21141 recovery working face at Qianqiu Coal Mine is used as an example, with data from the Canadian ESG microseismic monitoring system analyzed. The data time range is from 00:00 on April 1, 2010, to 00:00 on August 31, 2010. The data was analyzed using the "Mine Microseismic Visualization Interactive Analysis System," a software developed by the research team. The estimated fracture angles of the strata are shown in Fig 5. In the figure, the horizontal axis is the position of the 21141 working face lower drift, with the upper drift direction as positive; the vertical axis represents the depth coordinates.

Fig 5(a) shows the distribution of microseismic events with energy less than $10^4$ J in the dip profile, resulting in the average position of the strata fracture line: using the −50 m dip line as the reference, the vertical distance is approximately 270 m, and the horizontal distance from the 21141 working face lower drift is 50 m. $\tan\alpha = 270/50 = 5.4$, so the strata fracture angle $\alpha = 79.5°$. Fig 5(b) shows the distribution of events with energy between $10^4$ J and $10^5$ J in the dip profile, resulting in the average position of the strata fracture line: the vertical distance is approximately 250 m, and the horizontal distance from the 21141 working face lower drift is 50 m. $\tan\alpha = 250/50 = 5$, so the strata fracture angle $\alpha = 78.7°$. Fig 5(c) shows the distribution of microseismic events with energy between $10^5$ J and $10^6$ J in the dip profile, resulting in the average position of the strata fracture line: the vertical height is approximately 185 m, and the horizontal distance from the 21141 working face lower drift is 50 m. $\tan\alpha = 185/50 = 3.7$, so the strata fracture angle $\alpha = 74.9°$. Fig 5(d) shows the distribution of microseismic events with energy greater than $10^6$ J in the dip profile, resulting in the average position of the strata fracture line: the vertical height is approximately 260 m, and the horizontal distance from the 21141 working face lower drift is 50 m. $\tan\alpha = 260/50 = 5.2$, so the strata fracture angle $\alpha = 79.1°$. In summary, the average strata fracture angle $\alpha = (79.5° + 78.7° + 74.9° + 79.1°)/4 \approx 78°$.

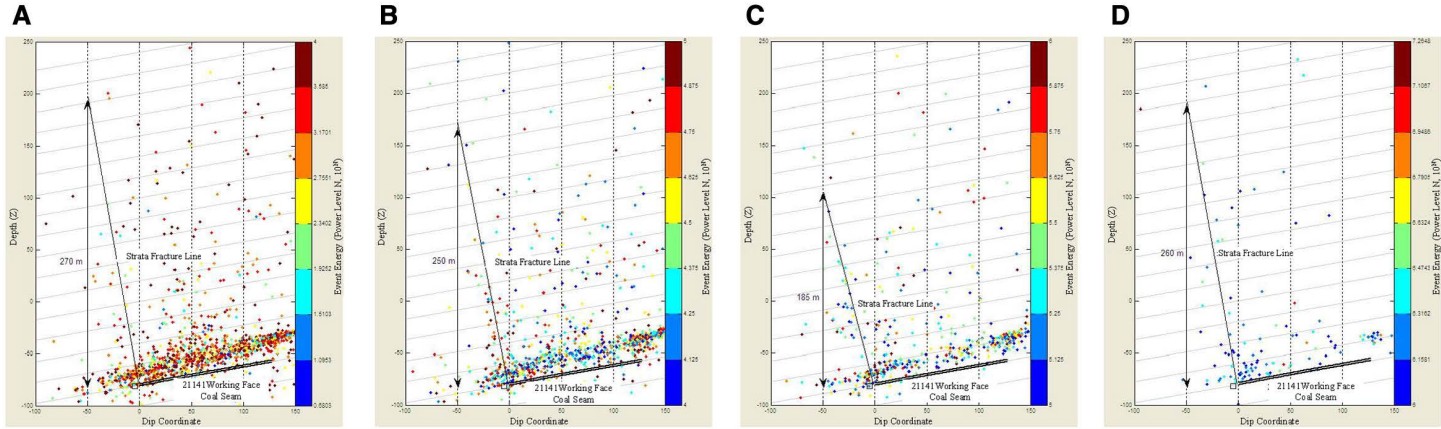

**Fig 5. Strata Fracture Angle Revealed by Microseismic Events [25]: (a) Energy less than $10^4$ J; (b) Energy from $10^4$ J to $10^5$ J; (c) Energy from $10^5$ J to 106 J; (d) Energy greater than 106 J.**

## 4.2. Strike span estimation method

By utilizing the high-precision positioning capabilities of mine microseismic monitoring, the strike span of the overlying strata in the recovery working face can be estimated. By setting the time, space (*i*-th layer key strata group), and intensity conditions, the distribution of microseismic events and their relative position to the working face can be used to estimate the strike span of the *i*-th layer key strata group. Using microseismic events from the 21141 working face at Qianqiu Coal Mine as the analysis subject, the real-time strike projection of the microseismic monitoring data was obtained through the software, allowing for the estimation of the key strata group's strike span. The specific analysis process is as follows: (1) Selection of Microseismic Event Time Range: the time range from 00:00:00 on January 1, 2011, to 00:00:00 on November 1, 2011, was selected for the analysis of the key strata group span. (2) Determination of Key Strata Groups: Borehole stratigraphic columns in the 21141 working face, and surface borehole measurement data from the 21121 goaf, four key overlying strata groups were identified, with thicknesses of $M_1 = 135$ m, $M_2 = 100$ m, $M_3 = 114$ m, and $M_4 = 176$ m. The altitude coordinate ranges of the corresponding strata groups are $Z_1$ (−45, 183), $Z_2$ (183, 301), $Z_3$ (301, 408), and $Z_4$ (408, 553). (3) Span Analysis of Key Strata Groups: The microseismic analysis data time range was set from 00:00:00 on January 1, 2011, to 00:00:00 on November 1, 2011, with an energy range of all events. The dip spatial range was set to 60 m on each side of the 21141 working face's upper and lower drifts, and the strike spatial range was from 500 m to 1000 m from the cut-off. The vertical spatial range was determined based on the key strata layers.

The resulting real-time strike projection of microseismic events is shown in Fig 6. In the figure, the horizontal axis represents the strike coordinates of the working face, with the 0-point of the strike coordinate at the cut-off of the working face; the vertical axis represents the time coordinate, in the format yyyy-mm-dd T hh:mm:ss. The gray part in the figure represents unmined coal, and the white part represents the goaf. The boundary line between the gray and white areas represents the position of the working face's strike.

Span Analysis of the First Key Strata Group: In the fixed time, intensity, strike space, and dip space range, the vertical range of the first key strata is selected, i.e., the vertical coordinates are (−45, 183), and the real-time strike projection of the microseismic events is shown in Fig 6(a). After adding the fracture analysis auxiliary line of the key strata group behind the working face, the span of the first key strata group is found to be about 60 m, $L_1 = 60$m. (2) Span Analysis of the Second Key Strata Group: The vertical range of the second key strata is selected, i.e., the vertical coordinates are (183, 301), and the real-time strike projection of the microseismic events is shown in Fig 6(b). After adding the fracture analysis auxiliary line of the key strata group behind the working face, the span of the second key strata group is found to be about 110 m, $L_2 = 110$m. (3) Span Analysis of the Third Key Strata Group: The vertical range of the third key strata is selected, i.e., the vertical coordinates are (301, 408), and the real-time strike projection of the microseismic events is shown in Fig 6(c). No data is available after the working face, indicating that the third key strata have not fractured. The load is uniformly distributed over the underlying strata and coal body. (4) Span Analysis of the Fourth Key Strata Group: In the fixed time, intensity, strike space, and dip space range, the vertical range of the fourth key strata is selected, i.e., the vertical coordinates are (408, 553), and no data is available in the real-time strike projection. Therefore, it is concluded that the fourth key strata have not fractured, and the load is uniformly distributed over the underlying strata and coal body.

## 5. Estimation of support pressure in the mining area and validation of effectiveness

### 5.1. Estimation of dip support pressure

#### 5.1.1. Calculation results of estimation method.
Taking the 21221 advancing working face in the deep section of the 21st mining area at Qianqiu Coal Mine as an example, the estimation of support pressure is demonstrated. The design of the 21221 working face has an average mining depth of approximately 800 m, with a 5 m small coal pillar left between

the working face and the 21201 goaf area. The lower part is solid coal. The working face has a design strike length of 1500 m and a dip length of 180 m. The estimation parameters are: $I$ = 120 m, $\gamma$ = 2.61 t/m$^3$, $\alpha$ = 78°, $H_{max}$ = 800 m, $M_1$ = 200 m, $M_2$ = 200 m, $M_3$ = 200 m, $H_1$ = 220 m, $H_2$ = 420 m, $H_3$ = 620 m. The estimation result of the inclined support pressure $\sigma = \Delta\sigma + \sigma q = \sigma q + \sum_1^n \sigma i$ is established as:

**A**

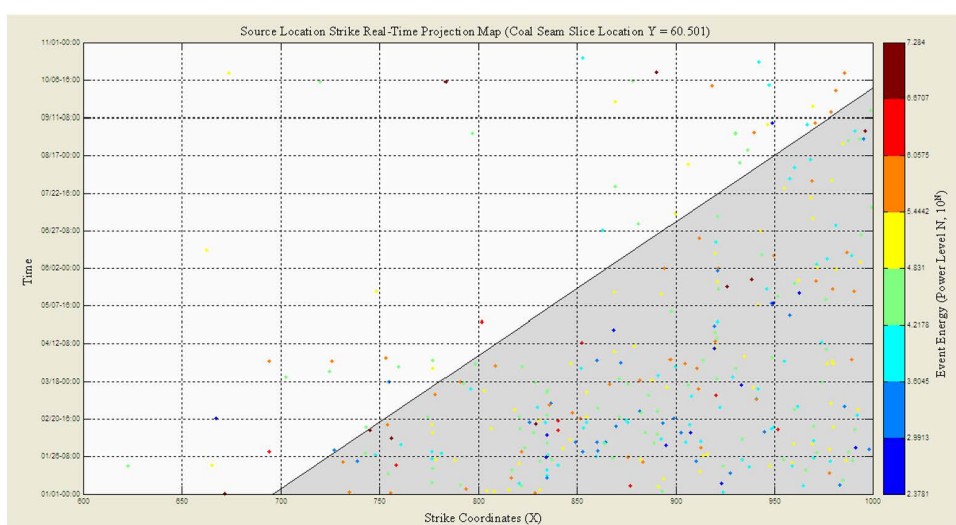

**B**

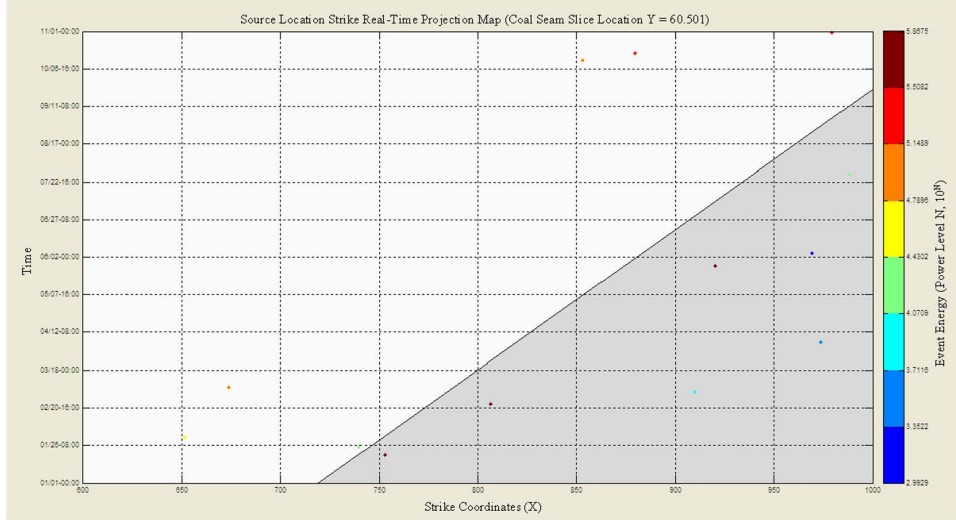

**Fig 6. Real-Time Strike Projection of Microseismic Events: (a) Projection of the First Key Strata Group's Range: Start and End Dates: 2011-01-01 00:00:00–2011-11-01 00:00:00; Event Energy (10$^N$ Joules) Power Level N Range: 2.3781 ~7.284; Number of Events: 319; Cumulative Mining Advance: 695.6~1033.95 meters; (b) Projection of the Second Key Strata Group's Range: Start and End Dates: 2011-01-01 00:00:00–2011-11-01 00:00:00; Event Energy (10$^N$ Joules) Power Level N Range: 2.9929~5.8675; Number of Events: 13; Cumulative Mining Advance: 719.25~1041.25 meters; (c) Projection of the Third Key Strata Group's Range: Start and End Dates: 2011-01-01 00:00:00–2011-11-01 00:00:00; Event Energy (10$^N$ Joules) Power Level N Range: 4.3722~5.1268; Number of Events: 4; Cumulative Mining Advance: 715.7~788.15 meters.**

C

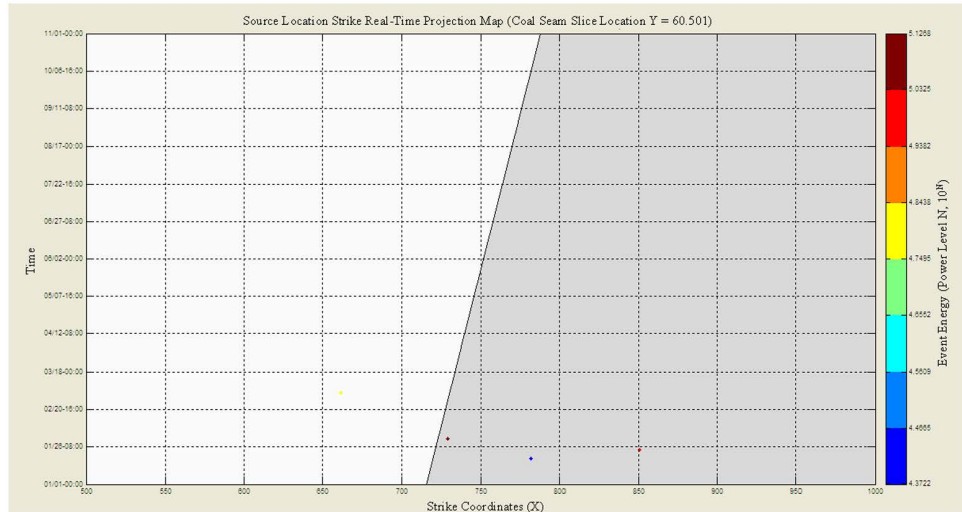

**Fig 6.** Continued.

$$
\sigma\left(x\right)=\begin{cases}
3.13+0.6126x & x \in [0, 26] \\
0.7352x & x \in [26, \mathbf{47}] \\
37.22-0.0648x & x \in [47, 89] \\
61.68-0.3388x & x \in [89, 94] \\
24.46+0.0612x & x \in [94, 132] \\
44.4-0.09x & x \in [132, 170] \\
64.4-0.2126x & x \in [170, 178] \\
39.94-0.0756x & x \in [178, 264] \\
20\text{MPa} & x \in [264, \infty)
\end{cases},
$$

(9)

For the synthetic nodes 0, 26 m, 47 m, 89 m, 94 m, 132 m, 170 m, 178 m, 264 m, ∞, the stress values at each node can be obtained by substituting into equation (9), as shown in Table 2.

**5.1.2. Validation of numerical simulation method.** Combining the estimated geological conditions and mining conditions at the location, numerical simulation of the excavation process of the 21181 and 21201 working faces was conducted using UDEC, with the results shown in Fig 7. When the distance from the lower drift of the 21201 working face to the coal rib is 90 m, the peak transmitted stress from the massive gravel rock reaches 29.4 Mpa. After that, as the distance from the lower drift of the 21201 working face increases, the transmitted stress from the massive gravel rock gradually decreases, eventually stabilizing at a distance of 210 m. The numerical calculation results are close to the stress estimation results. A comparison between the numerical calculation stress values and the stress estimation results is shown in Table 3.

**Table 2. Estimated Dip Support Pressure Results in the 21221 Working Face.**

| Distance from Goaf (m) | 0 | 26 | 47 | 89 | 94 | 132 | 170 | 178 | 264 | ∞ |
|---|---|---|---|---|---|---|---|---|---|---|
| Estimated Stress (MPa) | 3.13 | 19 | 34.55 | 31.45 | 29.83 | 32.08 | 29.1 | 26.55 | 20 | 20 |

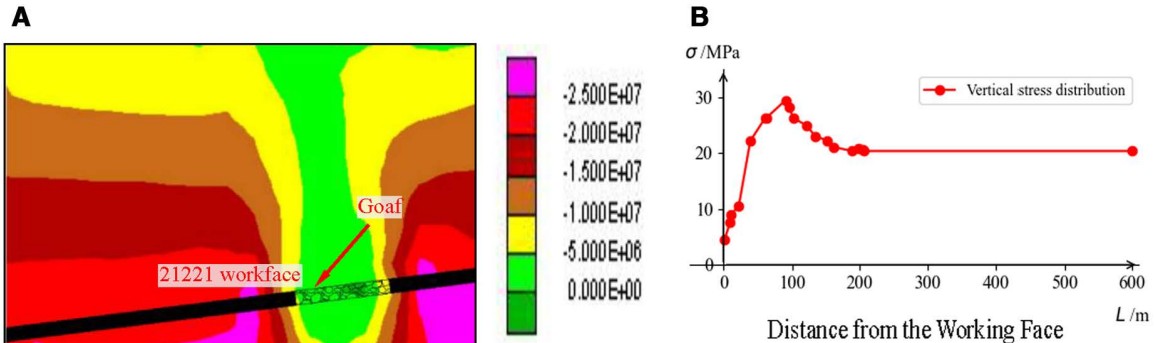

**Fig 7. Validation of Estimated Dip Support Pressure Using Numerical Calculation Results: (a) Stress Contour Map; (b) Numerical Calculation Stress Curve.**

**Table 3. Comparison of Estimated Dip Support Pressure Results in the 21221 Working Face with UDEC Simulation Results.**

| Distance from Goaf (m) | 0 | 26 | 47 | 89 | 94 | 132 | 170 | 178 | 264 | ∞ |
|---|---|---|---|---|---|---|---|---|---|---|
| Estimated Stress (MPa) | 3.13 | 19 | 34.55 | 31.45 | 29.83 | 32.08 | 29.1 | 26.55 | 20 | 20 |
| UDEC Distance (m) | 0 | 30 | 50 | 90 | 90 | 130 | 160 | 180 | 210 | ∞ |
| UDEC Stress (MPa) | 4.3 | 17.3 | 24 | 29.4 | 29.4 | 23.5 | 21.4 | 21 | 20.3 | 20.3 |

## 5.2. Estimation of strike support pressure

**5.2.1. Calculation results of estimation method.** Taking the 21141 working face in the western wing of the 21st mining area at Qianqiu Coal Mine in the Yima Coalfield as an example, the estimation of strike support pressure is demonstrated. The 21141 working face has an average mining depth of approximately 680 m, with a design strike length of about 1450 m and a dip length of 130 m. Based on the comprehensive stratigraphic column and microseismic monitoring data analysis, the estimation parameters are determined as: $\gamma = 2.61$ t/m³, $\alpha = 78°$, $L_{w1} = 60$ m, $L_{w2} = 110$ m, $h_{max} = 200$ m, $M_1 = 136$ m, $M_2 = 100$ m, $M_3 = 290$ m; $H_1 = 268$ m, $H_2 = 318$ m, $H_3 = 463$ m, $H_{max} = 680$ m.

The third key strata group and the fourth key strata group, based on on-site measurements, have not fractured and are treated as constant static loads acting on the underlying strata. Therefore, the self-weight of the unfractured strata overlying the goaf is approximated as half of the weight transferred forward to the working face, and the transmitted stress increment is related to the accumulated mining advance. In the estimation example, the maximum mining advance is used for estimation, and the microseismic analysis data time range is for locations more than 500 m from the track-side lower drift. The detailed calculation process (segmented synthesis $\sigma = \Delta\sigma + \sigma q = \sigma q + \sum_{1}^{n} \sigma i$) is as follows:

$$\sigma(x) = \begin{cases} 5.22 + 0.436x & x \in [0, 43] \\ 0.536x & x \in [43, 57] \\ 7.28 + 0.435x & x \in [57, 68] \\ 14.12 + 0.33667x & x \in [68, 98] \\ 77.4 - 0.309x & x \in [98, 114] \\ 70.12 - 0.245x & x \in [114, 135] \\ 63.28 - 0.194x & x \in [135, 145] \\ 81.028 - 0.321x & x \in [145, 197] \\ 17.748 MPa & x \in [197, \infty) \end{cases},$$

(10)

**Table 4. Estimated Strike Support Pressure at the 21141 Working Face Stopping Line Position.**

| Distance from Stopping Line (m) | 0 | 43 | 57 | 68 | 98 | 114 | 135 | 145 | 197 | ∞ |
|---|---|---|---|---|---|---|---|---|---|---|
| Estimated Stress (MPa) | 5.22 | 24 | 32.1 | 36.9 | 47.1 | 42.2 | 37.1 | 35.2 | 17.8 | 17.7 |

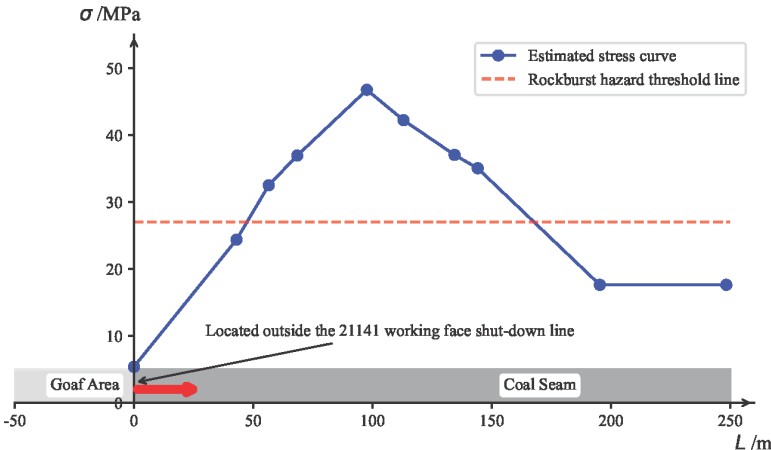

**Fig 8. Estimated Strike Support Pressure Results for the 21141 Working Face.**

By substituting the synthetic nodes and performing calculations, the estimated stress values for the synthetic nodes can be obtained, as shown in Table 4.

Using the stopping line of the 21141 working face as the 0-point on the horizontal axis, with the strike direction of the working face downward being the positive direction, the vertical axis represents the estimated stress values. A coordinate system is established, as shown in Fig 8.

**5.2.2. Validation of numerical simulation method.** Based on the estimated geological conditions and mining conditions at the location, UDEC numerical calculation software was used to simulate the distribution of vertical stress and the strata collapse and delamination in front of the working face after mining. The results are shown in Fig 9.

The estimated strike support pressure results for the 21141 working face stopping line position and the UDEC numerical calculation stress values are shown in Table 5. The stress estimation results are close to the numerical simulation results, but there is a significant difference in peak stress, approximately 13 MPa. However, when the working face is 500 m from the track-side downward drift, the estimated stress values are closer to the numerical calculation results. Considering that the microseismic analysis data for the working face is from locations near the 500 m distance from the track-side downward drift, it can be preliminarily concluded that the reliability of the estimation results is relatively high.

## 6. On-site application and validation

Based on the overlying strata spatial structure theory, an analysis of the microseismic monitoring data from the 21141 recovery working face was conducted to obtain the spatial fracture range of the overlying strata. Combining the mining conditions of the working face and the strata structure in the mining area, a spatial structure model of the overlying strata was established, as shown in Fig 10. Through a comprehensive analysis of surface subsidence observation data and microseismic monitoring results, it was concluded that the overlying massive hard gravel rock in the model exhibits delamination but has not fractured, with the spatial structure resembling an "unequal double-hole bridge". The bridge holes represent the goaf of the working face, and the bridge piers are the coal pillars to be mined. The wedge-shaped massive

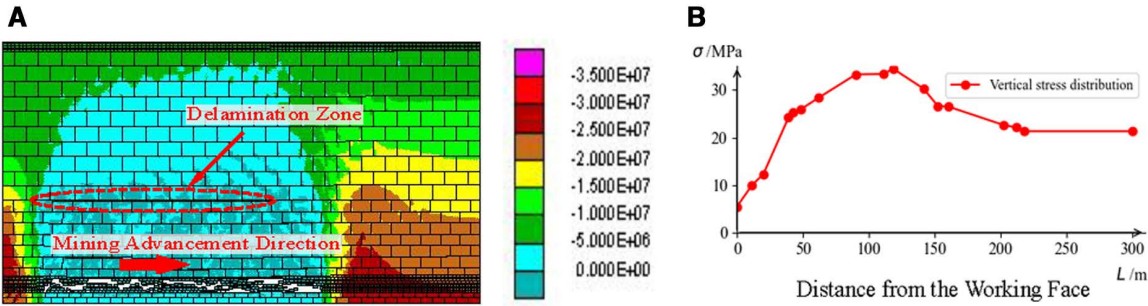

**Fig 9. UDEC Numerical Simulation Calculation: (a) Strike Vertical Stress Contour Map; (b) Strike Vertical Stress Curve.**

**Table 5. Comparison of Estimated Strike Support Pressure Results and UDEC Numerical Calculation Results for the 21141 Working Face.**

| Distance from the Stopping Line (m) | 0 | 43 | 57 | 68 | 98 | 114 | 135 | 145 | 197 | ∞ |
|---|---|---|---|---|---|---|---|---|---|---|
| Estimated Stress Value (MPa) | 5.2 | 24 | 32.1 | 36.9 | 47.1 | 42.2 | 37.1 | 35.2 | 17.7 | 17.7 |
| Distance from 500 m Below the Track Drift | 5.2 | 22.4 | 30 | 33.5 | 40.3 | 35.9 | 32 | 30.8 | 17.7 | 17.7 |
| UDEC Distance (m) | 0 | 40 | 60 | 70 | 100 | 111 | 130 | 140 | 200 | ∞ |
| UDEC Stress Value (MPa) | 5.1 | 24.6 | 28.3 | 29.7 | 33.7 | 34 | 32.5 | 30.4 | 22.7 | 21 |

gravel rock layer covers coal pillars (bridge piers) of varying widths. The 21141 working face, 21161 working face, and deep coal body are subjected to the combined stress of the overlying massive gravel rock self-weight, transmitted stress from the goaf, and tectonic stress. As a result, the 21141 working face, 21161 working face, and deep coal body (21221 working face) are long-term subjected to a high-stress field.

To explore the distribution of the three load zones in the mining area and the reasons for the small changes in surface subsidence data, Qianqiu Coal Mine conducted surface drilling above the 21121 working face goaf in 2010. The drilling was used to investigate the three load zones of the overlying strata above the goaf. Based on the integrity of the core samples and the phenomena observed during the drilling process, such as core loss, gas outbursts, water leakage, and slurry leakage, the range of the three load zones of the overlying strata from the surface to the goaf was inferred.

The results show that the grouping of the moving strata observed in the surface borehole is consistent with the overlying strata spatial structure model shown in Fig 10. The transmitted stress impact range caused by the overlying massive gravel rock in the goaf is quite large. Therefore, a lateral static support pressure estimation model needs to be established to estimate the transmitted stress, providing a basis for the working face's mining design or quantitative assessment of coal rockburst risk.

## 6.1. Quantitative coal burst risk evaluation based on superimposed stress estimation

### 6.1.1. Estimation of self-weight and transmitted stress from massive gravel rock. According to the overview of the 21141 working face, the average mining depth is 680 m. The average coal seam thickness is 25.2 m, with an average dip angle of 12°. The fracture angle of the strata is 78°, and the average mining depth of the 21121 working face is 650 m. Based on the surface borehole measurement results from the 21121 working face goaf area, and the comprehensive borehole column chart, the following calculation parameters can be determined: $I = 125$ m, $\gamma = 2.61$ t/m³, $\alpha = 78°$, $H_{max} = 680$ m, $M_1 = 136$ m, $M_2 = 100$ m, $M_3 = 290$ m, $H_1 = 193$ m, $H_2 = 311$ m, $H_3 = 506$ m, tan 78° = 4.7, cot 78° = 0.2126. Based on the geological conditions, mining technical conditions, and surface borehole measurement data analysis for the 21121 working face, and by analogy with the 21141 working face, microseismic measurement data was used to analyze the

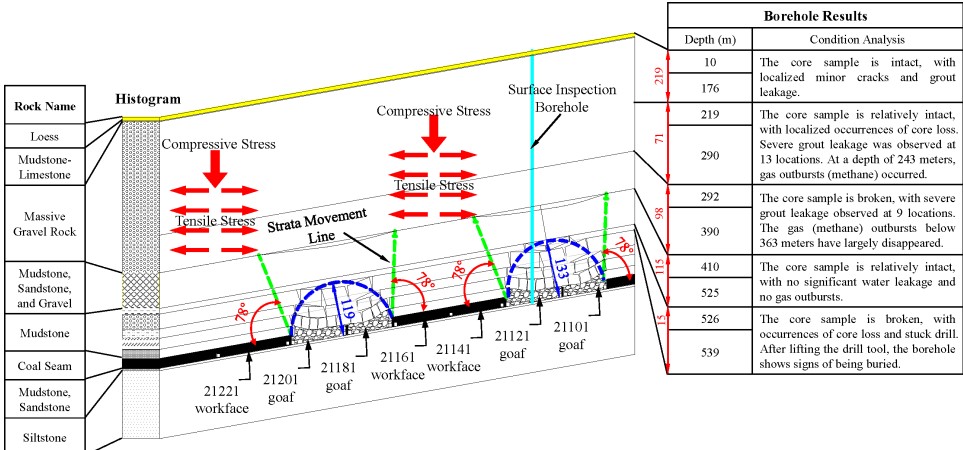

| Borehole Results | |
| --- | --- |
| Depth (m) | Condition Analysis |
| 10 | The core sample is intact, with localized minor cracks and grout leakage. |
| 176 | |
| 219 | The core sample is relatively intact, with localized occurrences of core loss. Severe grout leakage was observed at 13 locations. At a depth of 243 meters, gas outbursts (methane) occurred. |
| 290 | |
| 292 | The core sample is broken, with severe grout leakage observed at 9 locations. The gas (methane) outbursts below 363 meters have largely disappeared. |
| 390 | |
| 410 | The core sample is relatively intact, with no significant water leakage and no gas outbursts. |
| 525 | |
| 526 | The core sample is broken, with occurrences of core loss and stuck drill. After lifting the drill tool, the borehole shows signs of being buried. |
| 539 | |

**Fig 10. Correspondence Between the Western Wing of the Mining Area's Overlying Strata Spatial Structure and Surface Borehole Data.** The left side displays the lithology of the rock strata, while the right side presents the borehole depth and corresponding lithological condition analysis.

fracture angle of the strata. The transmitted stress from the massive gravel rock over the goaf of the 21121 and 21101 working faces was estimated. The estimated results are shown in Fig 11, where the X-axis of the coordinate system represents the dip distance, with the 0 point at the lower side of the 21121 working face undercut, and the Y-axis represents the estimated pressure value.

**6.1.2. Estimation of tectonic stress increment from $F_{3\text{-}7}$ Fault, $F_{3\text{-}9}$ Fault, and $F_{16}$ Fault.** The 21141 working face is affected by the $F_{3\text{-}7}$, $F_{3\text{-}9}$, and $F_{16}$ faults. Based on on-site engineering experience, the expected stress concentration factors and stress distribution ranges for each fault are shown in Table 6.

The stress distribution function near each fault can be represented by a piecewise function. The fault function expression for the upper side of the working face is shown in equation (11), and the fault function expressions for the lower side of the working face are shown in equations (12) and (13).

$$\sigma_{F_{3-9}}(x) = \begin{cases} 0.288 \times (x-230)+18 & x \in (230, 242.5] \\ 32.4-0.288 \times (x-242.5) & x \in (242.5, 255] \\ 0.288 \times (x-255)+18 & x \in (255, 267.5] \\ 32.4-0.288 \times (x-267.5) & x \in (267.5, 280] \end{cases},$$

(11)

$$\sigma_{F_{3-7}}(x) = \begin{cases} 0.51(x-1440)+18 & x \in (1440, 1447] \\ 21.6-0.51(x-1447) & x \in (1447, 1454] \\ 0.51(x-1454)+18 & x \in (1454, 1461] \\ 21.6-0.51(x-1461) & x \in (1461, 1468] \end{cases},$$

(12)

$$\sigma_{F_{16}}(x) = 9-0.045x \quad x \in (0, 200],$$

(13)

**6.1.3. Estimation of mining disturbance stress increment for the working face.** Based on experience and the general law of mine pressure distribution, the mining disturbance stress concentration factor caused by the mining of the 21141 working face is estimated as $k=1.5$, thus the mining disturbance stress is calculated, $\sigma_y=27$MPa.

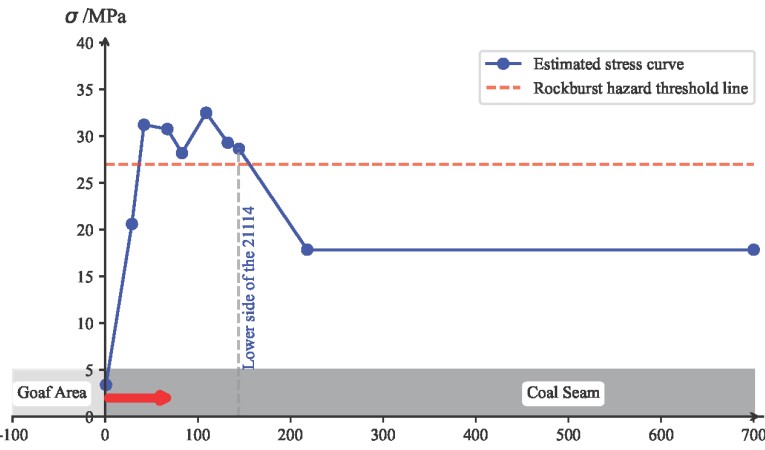

**Fig 11. Estimated Transmitted Stress Curve from the Overlying Massive Gravel Rock in the 21141 Working Face.**

**Table 6. Engineering Experience Table for Fault Stress Concentration Factors and Impact Ranges.**

| Fault Name | Fault Displacement (m) | Stress Concentration Factor (k) | Single-Sided Impact Range (m) |
|---|---|---|---|
| $F_{3-7}$ | 0~5 | 1.2 | 14 |
| $F_{3-9}$ | 0~32 | 1.4 | 25 |
| $F_{16}$ | >30 | 1.5 | 100 |

## 6.2. Evaluation conclusion of coal burst risk for the 21141 working face

Using the macro evaluation method, multi-factor coupling evaluation method, and the coal burst risk evaluation method based on superimposed stress estimation (the method presented in this paper), the coal burst risk area and risk level for the 21141 working face were evaluated and classified, followed by a comparative analysis of the effectiveness of the method proposed in this paper. Due to space limitations, the following description focuses on the multi-factor coupling evaluation method and the method presented in this paper.

**6.2.1. Multi-factor coupling evaluation.** For coal and rock bodies with a tendency for coal bursts, there are many factors influencing whether coal bursts will occur. However, the main influencing factors include coal and rock structure, geological structure, overlying strata spatial structure, tunnel surrounding rock structure, and mining depth. Based on the overlying strata spatial theory and mine pressure theory analysis, when the working face advances to areas near the first roof pressure, faults, working face advancement, and coal pillars, dynamic pressure is high, making it more likely to trigger coal bursts.

According to the multi-factor coupling evaluation method for coal bursts, the coal and rock structure of the working face, geological structure, overlying strata spatial structure, and tunnel surrounding rock structure are separately defined as hazardous areas. Then, the coal burst risk areas caused by each factor are superimposed and coupled to define the final coal burst risk area and its risk level. After multi-factor coupling evaluation, the coal burst risk area for the 21141 working face is divided as shown in Fig 12.

The green area represents the weak coal burst risk zone, the yellow area represents the moderate coal burst risk zone, and the red area represents the strong coal burst risk zone. Under the influence of the strata transmitted stress formed by the overlying strata spatial structure, the entire lower side of the 21141 working face undercut is within a high-stress zone. Based on the coal and rock structure and surrounding rock structure (with a bottom coal layer thicker than 2 meters), the entire lower side of the undercut is at moderate coal burst risk. When further coupled with the influence of working face

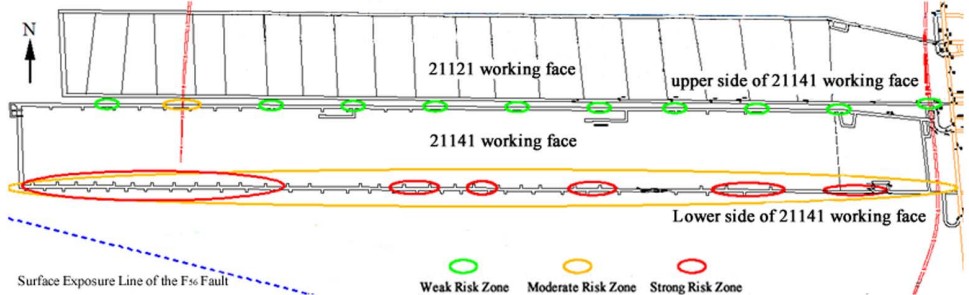

**Fig 12. Multi-Factor Coupling Evaluation Conclusion of Coal Burst Risk for the 21141 Working Face.** As illustrated in the figure, the color-coded scheme represents varying rockburst risk levels: green (low risk), yellow (moderate risk), and red (high risk).

advancement, tectonic stress, as well as special facilities such as water reservoirs and refuge chambers, 6 locations along the lower side of the working face are identified as strong coal burst risk zones.

**6.2.2. Superimposed stress estimation and coal burst risk area division.** Based on the superimposed stress estimation results from self-weight stress, transmitted stress from massive gravel rock, tectonic stress, and mining disturbance stress, as shown in Fig 13, the dynamic stress ratio $d = \sigma/[\sigma_c]$ is used as the risk evaluation standard. The empirical $d$ values for coal burst risk area division are: no risk $d \in (0, 1.5)$, weak risk $d \in [1.5, 1.8)$, moderate risk $d \in [1.8, 2]$, and strong risk $d \in (2, +\infty)$. Therefore, based on the superimposed stress estimation results and the $d$ value, the evaluation conclusion is as follows: The upper side of the 21141 working face undercut has no coal burst risk, while the entire lower side of the 21141 working face undercut is classified as a strong coal burst risk zone.

**6.2.3. Comparison of evaluation results from multiple methods.** For ease of observation, a comparative analysis of the evaluation process and results from the macro evaluation method, multi-factor coupling evaluation method, and the method presented in this paper was conducted. The results are shown in Table 7.

From the evaluation results, by combining the macro evaluation results, multi-factor coupling evaluation results, and the quantitative evaluation results based on superimposed stress estimation, it can be concluded that the 21141 working face has coal burst risk. Among these, the upper side of the 21141 working face, near faults F3-9 and F3-7, shows weak coal burst risk, while the lower side of the working face presents strong coal burst risk. In particular, areas near the cutting eye, water reservoir, and fault F3-7 show high stress concentration. Additionally, the working face advancement region experiences intense disturbances from overlying strata movement, making these areas the highest risk for coal bursts.

Upon reviewing the field pressure relief construction records for the 21141 working face, it was found that large diameter $\Phi 120\,mm$ pressure relief boreholes were drilled on both sides of the tunnel. The pressure relief borehole construction in the lower side of the warning area showed fewer dynamic phenomena, whereas the pressure relief process in the upper side was accompanied by frequent coal blasts. The 120 mm pressure relief borehole, drilled to a depth of 25 m, produced coal powder sufficient to fill two mining carts, fully proving that the area had reached the red warning level for coal burst. This aligns with the analysis above. On November 3, 2011, at 19:18, a coal burst occurred in the lower side of the 21221 excavation working face at Qianqiu Coal Mine, with a microseismic event energy of $3.5 \times 10^8$ J. This is consistent with the analysis above.

In summary, from the results in Table 7, it can be seen that in the macro evaluation method, the comprehensive index method combines laboratory results of coal samples and can qualitatively analyze coal burst risks in local areas, but it cannot overcome the limitations of sampling points. The index diagnostic method requires subjective scoring by evaluators to obtain evaluation results. These methods have significant subjectivity and are difficult to implement for coal burst risk area classification. The multi-factor coupling evaluation method has improvements, allowing for detailed classification of risk levels in different areas of the working face, but the scoring of evaluation indicators is still subjective. The method proposed in this paper (the stress superposition method) takes a quantitative approach, considering structural influences,

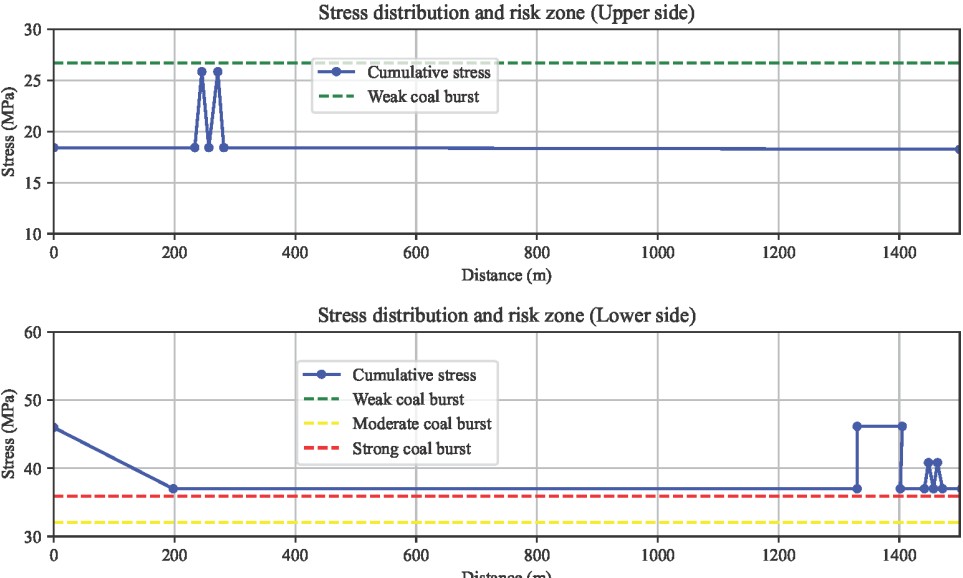

**Fig 13. Superimposed Estimation of Support Pressure and Coal Burst Risk Area Division for the 21141 Working Face Under the Longwall.**

**Table 7. Comparison of Coal Burst Risk Evaluation Results for the 21141 Working Face Based on Multiple Methods.**

| Evaluation Method | Specific Method | Evaluation Indicators | Evaluation Type | Evaluation Results |
|---|---|---|---|---|
| Macro Evaluation | Comprehensive Index Method | Dynamic failure time, impact energy index, elastic energy index, etc., coal sample impact tendency index. | Qualitative Evaluation | Coal burst may occur |
| | Impact Possibility Index Diagnostic Method | Mining depth, coal seam impact tendency, hard roof strata, tectonic stress concentration, and other coal burst influencing factors. | Qualitative Evaluation | High coal burst risk |
| Multi-factor Coupling Evaluation | —— | Coal and rock structure, geological structure, overlying strata spatial structure, tunnel surrounding rock structure, and mining depth. | Qualitative Evaluation | Weak coal burst risk in multiple locations on the upper side of the 21141 working face, some moderate risk; the entire lower side of the working face has moderate coal burst risk, with local high-risk areas. |
| This Method (Paper's Method) | —— | Stress-based, with consideration of mining disturbances, etc. | Quantitative Evaluation | No coal burst risk on the upper side of the 21141 working face; the entire lower side is a high coal burst risk area. |

mining disturbances, and more, further enhancing the reliability of stress estimation. However, this method only considers the stress aspect, and it has certain limitations. Therefore, using all three methods together to build a qualitative + quantitative combined evaluation method can yield better results.

## 7. Discussion and conclusions

### 7.1. Discussion

For mining/advancing working faces that have already formed a system, an accurate pre-evaluation of coal burst risk areas and risk levels can greatly enhance the effectiveness of coal burst prevention and control. Chinese experts and

scholars have conducted extensive research and field practices on coal burst risk evaluation methods for working faces, achieving a significant amount of research. However, these evaluation methods are mostly qualitative, and a more quantitative coal burst risk evaluation would be more meaningful for on-site applications. Currently, the most commonly used coal burst evaluation methods in coal mines are macro evaluation methods and multi-factor coupling evaluation methods. Based on previous research, this paper proposes a pre-evaluation method for coal burst risk based on superimposed stress estimation to improve the accuracy of coal burst risk evaluation for working faces in extremely thick coal seams under massive gravel rock: The method is based on self-weight stress calculation, transmitted stress estimation from massive gravel rock, tectonic stress quantitative analysis, and mining disturbance stress estimation. A planar rectangular coordinate system is established, and the superimposed stress estimation method is applied to quantitatively estimate the total superimposed stress in the mining area. Using the dynamic stress ratio ($d$) as the coal burst risk classification standard, the risk area and risk level are divided within the coordinate system. These results are then mapped to the pre-evaluated working face, yielding a more quantitative coal burst risk evaluation conclusion. The conclusions obtained from integrating the macro evaluation method and multi-factor coupling evaluation method are primarily quantitative, with qualitative evaluation supplementing it, and the risk area and risk level are comprehensively determined.

The existing coal burst risk assessment methods mainly focus on analyzing the physical and mechanical properties of the coal seam itself, mining disturbances, and stress distribution. These methods often concentrate on the coal seam itself while neglecting the impact of the overlying rock layers (such as ultra-thick conglomerates), which can lead to inaccurate results in specific scenarios. In contrast, the method proposed in this paper combines the stress superposition effect of ultra-thick coal seams under ultra-thick conglomerates, utilizing a combination of microseismic monitoring and UDEC numerical simulation techniques. Through "stress superposition estimation," it provides a quantitative assessment of rockburst risk. In specific scenarios, especially when facing the special cover conditions of ultra-thick conglomerates, this method can more accurately simulate the stress interactions between the coal seam and the overlying rock layer, overcoming the shortcomings of traditional methods under complex geological conditions. Specifically, the advantages are: (1) it integrates stress calculations, microseismic monitoring, and numerical simulations to comprehensively evaluate the interactions between the coal seam and overlying rock layers, providing a more thorough risk assessment; (2) the "stress superposition estimation" method allows for accurate estimation of the stress distribution and accumulation at different layers, enhancing the predictive ability for rockburst risk. In summary, the quantitative assessment method and results of impact risk evaluation are urgently needed in engineering practice. They not only guide on-site rockburst pressure relief and hazard mitigation, improving relief effectiveness and reducing construction costs but also contribute to better safety management.

The external superimposed total stress and the rockburst types in the thick coal seam roadway surrounding rock are the core research areas in understanding the mechanism of rockbursts under ultra-thick conglomerates. This paper explores valuable aspects of ultra-thick conglomerate stress transfer estimation, the mechanism of rockburst occurrence in ultra-thick coal seams, and monitoring, early warning, and control technologies for rockbursts, proposing some new perspectives. However, several points still require further in-depth research, including:

(1) The structural stress of geological structures such as reverse faults and transformation zones cannot yet be calculated accurately and rigorously, and it is not possible to perform a quantitative calculation of structural stresses. Only semi-quantitative estimation has been conducted. The current quantitative calculation methods for structural stresses have limitations mainly due to the simplified errors of empirical assumptions, the roughness of parameter values, and the lack of dynamic effects. Despite the limitations in terms of accuracy, this method's advantage lies in its ability to quickly estimate structural stresses by simplifying the process empirically, which remains a distinct strength.

(2) The static support pressure estimation model for dip/strike presented in this paper has certain limitations. These limitations mainly manifest in the following ways: First, in terms of load distribution, the model ignores the

rheological effect of weak interlayers and does not consider local load redistribution caused by the formation of plastic zones in the coal seam; the model also fails to account for significant load eccentricity caused by extreme dip angles. Second, in the boundary condition assumptions, it does not consider the stepped fractures that occur when the conglomerate thickness changes abruptly (which requires segment angle corrections), the time effects of the fracture angle under dynamic mining conditions, and the dynamic disturbances caused by mining speed. Third, in terms of rock layer behavior, the anisotropic effects caused by dominant joint sets need to be considered.

(3) Furthermore, microseismic positioning accuracy is critical. The error propagation chain follows "microseismic error → stress calculation → risk assessment". Positioning inaccuracies lead to erroneous determinations of fracture angles ($a$) and strike spans ($L$), which subsequently cause miscalculations in stress distribution. These stress computation errors ultimately result in unreliable risk evaluations. To address this, we implemented a dual-path optimization strategy for microseismic data processing: ① Sensor-level enhancements: Upgraded localization algorithms using double-difference tomography; ② Data-processing refinements: Introduced DBSCAN clustering analysis. Developed waveform cross-correlation filters. These measures collectively enhanced positioning precision from ±8.3m to ±3.7m (verified by blast tests), effectively breaking the error propagation chain.

## 7.2. Conclusion

Taking a typical rockburst-prone coal mine with ultra-thick coal seams beneath massive conglomerate strata in China as the engineering background, this study focused on two core issues: the estimation of impact energy sources and the quantitative assessment of rockburst risk. A comprehensive theoretical investigation and field validation were conducted. The main conclusions are as follows:

(1) Based on an in-depth analysis of the rockburst characteristics under the geological conditions of massive conglomerate overlying ultra-thick coal seams, and considering the structural morphology of the overburden, this study investigated stress transmission estimation through conglomerates and quantitative evaluation of tectonic stress (including faults and phase transition zones). A stress superposition estimation method integrating multiple stress sources was proposed, leading to the development of a rapid estimation model for abutment pressure under massive conglomerate conditions. This model revealed the stress transmission pattern in such geological settings.

(2) Under conditions of insufficient overburden movement, a novel method was proposed to estimate the fracture angle and strike-span based on microseismic profile projection. Taking the 21141 working face in Qianqiu Coal Mine as an example, microseismic data from the ESG monitoring system revealed that the fracture angle in the underlying roadway was 78°, and the spans of the first and second key strata were 60 m and 110 m, respectively, confirming the practicality and accuracy of the method.

(3) Comparative analysis between UDEC numerical simulations and microseismic monitoring data showed a significant correlation between abutment pressure distribution and high-frequency microseismic event zones. This result validated the reliability of both the abutment pressure estimation and the stress superposition models, reinforcing the theoretical foundation of this study.

(4) Based on the proposed models for rapid estimation of strike and dip abutment pressures, the impact energy source in the 21141 working face was calculated. Furthermore, a quantitative risk evaluation and zoning of rockburst hazards was conducted using the stress superposition approach. A classification standard for risk levels was proposed, improving both the accuracy and practicality of rockburst prediction. This research provides theoretical support and engineering guidance for rockburst prevention under complex geological conditions.

## Author contributions

**Conceptualization:** Quande Wei, Guangyu Yang.

**Data curation:** Quande Wei, Guangyu Yang.

**Formal analysis:** Quande Wei, Dongsheng Jiang, Yi Liu, Qilin Hao.

**Funding acquisition:** Quande Wei, Quanjie Zhu.

**Investigation:** Quande Wei, Quanjie Zhu, Guangyu Yang, Yi Liu, Qilin Hao.

**Methodology:** Quande Wei, Quanjie Zhu, Guangyu Yang, Yi Liu, Qilin Hao.

**Project administration:** Quande Wei.

**Resources:** Quande Wei, Yi Liu, Qilin Hao.

**Software:** Quande Wei.

**Supervision:** Quande Wei.

**Validation:** Quande Wei, Guangyu Yang, Dongsheng Jiang, Yingnan Hao.

**Visualization:** Quanjie Zhu, Dongsheng Jiang, Yingnan Hao.

**Writing – original draft:** Quanjie Zhu, Dongsheng Jiang, Yingnan Hao.

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
