## [Decision Letter · Decision Letter 0]

24 Mar 2025

PONE-D-25-09536Study on Estimation Method for Coal Burst Force Sources in Extremely Thick Coal Seams and Quantitative Evaluation of Coal Burst RiskPLOS ONE

Dear Dr. Zhu,

Thank you for submitting your manuscript to PLOS ONE. After careful consideration, we feel that it has merit but does not fully meet PLOS ONE’s publication criteria as it currently stands. Therefore, we invite you to submit a revised version of the manuscript that addresses the points raised during the review process.

We look forward to receiving your revised manuscript.

Kind regards,

Kang Wang, Ph.D.

Academic Editor

PLOS ONE

“This work is financially supported by the Key Science and Technology Program of the Ministry of Emergency Management (2024EMST070702), the Natural Science Foundation of Hebei Province of China (E2023508021) and the Fundamental Research Funds for the Central University (3142024005, 3142021002).”

“This work is financially supported by the Key Science and Technology Program of the Ministry of Emergency Management (2024EMST070702), the Natural Science Foundation of Hebei Province of China (E2023508021) and the Fundamental Research Funds for the Central University (3142024005, 3142021002).’

“This work is financially supported by the Key Science and Technology Program of the Ministry of Emergency Management (2024EMST070702), the Natural Science Foundation of Hebei Province of China (E2023508021) and the Fundamental Research Funds for the Central University (3142024005, 3142021002).”

5. In this instance it seems there may be acceptable restrictions in place that prevent the public sharing of your minimal data. However, in line with our goal of ensuring long-term data availability to all interested researchers, PLOS’ Data Policy states that authors cannot be the sole named individuals responsible for ensuring data access (http://journals.plos.org/plosone/s/data-availability#loc-acceptable-data-sharing-methods).

Reviewers' comments:

Reviewer's Responses to Questions

**Comments to the Author**

1. Is the manuscript technically sound, and do the data support the conclusions?

Reviewer #1: Yes

Reviewer #2: Yes

Reviewer #3: Yes

Reviewer #4: No

2. Has the statistical analysis been performed appropriately and rigorously? 

Reviewer #1: Yes

Reviewer #2: Yes

Reviewer #3: I Don't Know

Reviewer #4: N/A

3. Have the authors made all data underlying the findings in their manuscript fully available?

Reviewer #1: Yes

Reviewer #2: Yes

Reviewer #3: Yes

Reviewer #4: Yes

4. Is the manuscript presented in an intelligible fashion and written in standard English?

Reviewer #1: Yes

Reviewer #2: Yes

Reviewer #3: Yes

Reviewer #4: Yes

5. Review Comments to the Author

Reviewer #1: Your manuscript presents a comprehensive study on estimating coal burst force sources in extremely thick coal seams beneath massive gravel rock. However, enhancing the clarity of the methodology, broadening the discussion's practical context, and providing further justification for certain methodological choices would strengthen the paper.

1、 Explicitly list the key assumptions made in the stress estimation models (e.g., load distribution assumptions, boundary conditions). This would help readers better understand the limitations and applicability of your models.

2、 While the discussion section touches on future research directions, a more detailed discussion on the limitations of the current approach (e.g., uncertainty in tectonic stress quantification) would be valuable.

3、 Consider adding more detailed captions and ensuring that all axes, scales, and legends are clearly explained.

4、 While the introduction explains the significance of the study, it could benefit from more in-depth references to recent research in the past five years. Highlight gaps in the literature this study addresses, such as: compgeo.2024.106095; tafmec.2024.104691; fuel.2023.129584

5、 Emphasize how the quantitative approach can bridge the gap between theoretical predictions and on-site applications in complex geological settings.

6、 Highlight avenues for future research, such as addressing the challenges of non-linear interactions between different stress sources.

7、 Strengthen your discussion by comparing your method not only with traditional qualitative approaches but also with emerging quantitative methods.

8、 The comparison with existing coal burst risk evaluation methods is a strength. Expanding this section to discuss potential scenarios where your method performs particularly well (or less well) compared to conventional methods could further highlight its advantages.

Reviewer #2: 1- The title should be improved.

2- The objectives and the rationale of the study are recommended to be clearly stated.

3- The concluding remarks of the abstract are not well-written. It's merely the repetition of the objectives and title of the manuscript. Please add method limitations and justification to the abstract.

4- The innovation of using this study is not very clear. I do not see a clear reason that this study can perform better than others. Why did the authors choose the method for this study?

5- The necessity & novelty of the manuscript should be presented and stressed in the "Introduction" section.

6- The application/theory/method/study reported is not in sufficient detail to allow for its replicability and/or reproducibility. Therefore, it is suggested to make it clear to show all steps to build the model.

7- The problem statement and gap study are not clear.

8- The method is not clear. Therefore, it must be shown and clarified to show all steps.

9- The interpretation of results and study conclusions are not supported by providing the reasons behind why they show that. Therefore, it is recommended to deepen the discussion.

10- It is recommended to emphasize the strengths of the study clearly.

11- The limitations of the study should be stated.

12- The manuscript structure, flow, or writing needs some improvements.

13- The manuscript is benefit from language editing. The English of the paper is readable; however, I would suggest the authors to have it checked preferably by a native English-speaking person to avoid any mistakes.

14- I noticed that the conclusion section tends to repeat the abstract and results. The conclusion paragraph should be short, impactful, and direct the reader to this research's next steps and opportunities.

15- It will be nice to add some new references to show that your study is updated, such as: Zhou, Zhanxin, and Ruibo Wu. "Stock Price Prediction Model Based on Convolutional Neural Networks." Journal of Industrial Engineering and Applied Science 2.4 (2024): 1-7; Alakbari, Fahd Saeed, et al. "Prediction of critical total drawdown in sand production from gas wells: Machine learning approach." The Canadian Journal of Chemical Engineering 101.5 (2023): 2493-2509 .; Alakbari, Fahd Saeed, et al. "Deep learning approach for robust prediction of reservoir bubble point pressure." ACS omega 6.33 (2021): 21499-21513 .; Alakbari, Fahd Saeed, et al. "A gated recurrent unit model to predict Poisson's ratio using deep learning." Journal of Rock Mechanics and Geotechnical Engineering 16.1 (2024): 123-135; Zhou, Zhanxin, and Ruibo Wu. "Stock Price Prediction Model Based on Convolutional Neural Networks." Journal of Industrial Engineering and Applied Science 2.4 (2024): 1-7; Wu, Ruibo, Tao Zhang, and Feng Xu. "Cross-Market Arbitrage Strategies Based on Deep Learning." Academic Journal of Sociology and Management 2.4 (2024): 20-26. Alakbari, Fahd Saeed, et al. "Prediction of critical total drawdown in sand production from gas wells: Machine learning approach." The Canadian Journal of Chemical Engineering 101.5 (2023): 2493-2509 . Hassan, Anas M., et al. "A new insight into smart water assisted foam SWAF technology in carbonate rocks using artificial neural networks ANNs." Offshore Technology Conference Asia. OTC, 2022.

Reviewer #3: Dear Authors,

Thank you for submitting your manuscript, "Study on Estimation Method for Coal Burst Force Sources in Extremely Thick Coal Seams and Quantitative Evaluation of Coal Burst Risk," to PLOS ONE. Your research tackles a pressing challenge in underground mining safety—preventing coal bursts in thick coal seams—and your work at Qianqiu Coal Mine in Henan, China, offers valuable insights. The combination of stress estimation, microseismic monitoring, numerical modeling, and field validation is impressive, and your focus on enhancing mine safety is both timely and impactful. To help strengthen your manuscript and maximize its contribution to the field, I’d like to offer some suggestions for improvement.

1. Introduction

1. What Are "Extremely Thick Coal Seams"?

In the introduction, please specify what "extremely thick" means in your study—perhaps a thickness range (e.g., >10 meters?)—so readers can better understand the context of your work.

2. Simplify Technical Descriptions

The sections on stress estimation and microseismic monitoring are detailed but dense. A simpler explanation or a concise summary of these methods would improve readability while keeping the technical depth intact.

2. Methodology

3. Quantifying Tectonic Stress

Tectonic stress plays a big role in coal bursts, but your study treats it in a semi-quantitative way. Could you include more precise measurements, perhaps using geophysical data or advanced models, to quantify its impact?

4. Why These Microseismic Parameters?

You use fracture angles and strike spans from microseismic data to assess coal burst risk. Could you explain why these specific parameters were chosen and how they predict bursts?

5. Comparing to Other Models

Your new risk evaluation model is exciting, but how does it stack up against existing tools like BurstRisk? A direct comparison would highlight what makes your approach stand out.

3. results

6. Your work fits into a global effort to improve mine safety. Comparing it to studies like the one on roadway rockbursts in deep coal seams (e.g., this study) would show how your research builds on or differs from what’s out there.

7. Since you’ve developed a quantitative risk evaluation method, explaining how it improves on semi-quantitative systems (like BurstRisk) would strengthen your case.

8. If possible, include datasets like microseismic readings or stress measurements as supplementary files. This would let others verify or build on your work.

9. If you used proprietary tools or data, consider suggesting open-source alternatives or other ways readers could replicate your results.

4. Practical Applications and Limitations

10. You mention pre-cracking massive gravel rock to reduce burst risks—could you give more details on practical techniques or tools that could make this happen?

11. Adding a section on your study’s limitations (e.g., assumptions in your stress models or challenges with tectonic stress) would give a fuller picture and point the way for future work.

5. Better Visuals

12. Diagrams like stress maps or risk zones would make your findings easier to grasp.

13. A table summarizing key results—say, how different stress factors contribute or how your predictions match real data—would help readers quickly see your main points.

Reviewer #4: 1. In the Introduction section, rockbursts phenomenon is described in general terms focusing on potential hazards. However, the rationale for the study is expressed through subjective judgments about the importance of the quantitative estimation approach/method. Instead of discussing the weaknesses of alternative approaches/methods, the authors focused on presenting the potential benefits of their proposed method. So noone will know about the rationale need for using quantitative estimation method.

Although the aim of the study is presented in sufficient detail, due to the lack of a discussion on the rationale makes Introduction can be accepted absent.

2. In the study frequent refereneces to previous/other reserches on the subject (e.g., “experts and scholars have conducted extensive research and field practices,” L573–574), these studies are not properly cited.

3. Some of the approaches used in calculating effective stresses appear to be based on general geometry-weight relationships (stress calcualtions), while others seem to be constructed based on the authors’ assumptions. The rationale for using these assumption-based formulations is not clearly stated. Consequently, due to the calculations they are based on, it can be argued that the study lacks a solid theoretical foundation.

4.The manuscript appears to be an attempt to merge two different and largely unrelated topics. The title, “Study on Estimation Method for Coal Burst Force Sources in Extremely Thick Coal Seams and Quantitative Evaluation,” suggests that the primary focus is on the estimation of coal burst forces. However, the Conclsuions section presents only the findings from a microseismic monitoring study. Thus, not only does the main theme lack a sound theoretical foundation and rely on hypothetical assumptions, but the microseismic monitoring work—although potentially worthy of a separate study—is only loosely related and is presented at a purely technical note level. Despite a limited connection, these are conceptually distinct topics that should be addressed separately.

Depending on the above reasons, it would be appropriate to reject this manuscript.

In addition, there are issues of carelessness in terms of formatting throughout the article.

Examples:

1. A subheading appears as: “1111111Stress Analysis of the F16 Reverse Fault and Phase Boundary Zone.”

2. There is insufficient spacing between tables/figures and the following paragraphs.

6. PLOS authors have the option to publish the peer review history of their article (what does this mean? ). If published, this will include your full peer review and any attached files.

**Do you want your identity to be public for this peer review?** For information about this choice, including consent withdrawal, please see our Privacy Policy .

Reviewer #1: No

Reviewer #2: No

Reviewer #3: No

Reviewer #4: No

---

## [Author Response · Author response to Decision Letter 1]

9 Apr 2025

Please refer to the attached file/document.

---

## [Decision Letter · Decision Letter 1]

30 Apr 2025

PONE-D-25-09536R1Estimation Method for Impact Force Source in Thick Coal Seams and Its Application in Rockburst Risk QuantificationPLOS ONE

Dear Dr. Zhu,

Thank you for submitting your manuscript to PLOS ONE. After careful consideration, we feel that it has merit but does not fully meet PLOS ONE’s publication criteria as it currently stands. Therefore, we invite you to submit a revised version of the manuscript that addresses the points raised during the review process.

We look forward to receiving your revised manuscript.

Kind regards,

Kang Wang, Ph.D.

Academic Editor

PLOS ONE

Journal Requirements:

Reviewers' comments:

Reviewer's Responses to Questions

**Comments to the Author**

1. If the authors have adequately addressed your comments raised in a previous round of review and you feel that this manuscript is now acceptable for publication, you may indicate that here to bypass the “Comments to the Author” section, enter your conflict of interest statement in the “Confidential to Editor” section, and submit your "Accept" recommendation.

Reviewer #1: All comments have been addressed

Reviewer #3: All comments have been addressed

2. Is the manuscript technically sound, and do the data support the conclusions?

Reviewer #1: Yes

Reviewer #3: Partly

3. Has the statistical analysis been performed appropriately and rigorously? 

Reviewer #1: Yes

Reviewer #3: Yes

4. Have the authors made all data underlying the findings in their manuscript fully available?

Reviewer #1: Yes

Reviewer #3: Yes

5. Is the manuscript presented in an intelligible fashion and written in standard English?

Reviewer #1: Yes

Reviewer #3: No

6. Review Comments to the Author

Reviewer #1: (No Response)

Reviewer #3: Paper Title: "Estimation Method for Impact Force Source in Thick Coal Seams and Its Application in Rockburst Risk Quantification"

1. Rewrite abstract’s conclusion to focus on key findings (e.g., better warnings) instead of repeating goals.

2. Method section is too complex with heavy math (e.g., equations 1–7). Add a simple diagram or step-by-step list to explain the model.

3. Clarify how microseismic parameters (fracture angle, strike span) directly link to rockburst risk, possibly with a picture.

4. Tectonic stress estimates (Section 3.2) are semi-quantitative and lack precision. Suggest specific tools (e.g., seismic tomography, in-situ stress measurements) for future work.

5. Note that assuming symmetric stress around faults oversimplifies things; mention potential fault variations (e.g., dip angle changes).

6. Compare the method to tools like BurstRisk in a table, showing when it works best or worst (e.g., in thinner seams or simpler mines).

7. Replace irrelevant references (e.g., Zhou & Wu, 2024 on stock prices) with recent rockburst or mining studies.

8. Improve figure captions (e.g., Fig. 4) to explain axes, scales, and why energy levels matter.

9. Add a table showing how much each stress type (gravel rock, faults, mining) contributes to total risk.

10. Specify which mine conditions (e.g., steep seams, high fault density) might limit the method’s effectiveness.

11. Explain how errors in microseismic data accuracy could impact results.

Recommendation: Minor Revisions Needed

7. PLOS authors have the option to publish the peer review history of their article (what does this mean? ). If published, this will include your full peer review and any attached files.

**Do you want your identity to be public for this peer review?** For information about this choice, including consent withdrawal, please see our Privacy Policy .

Reviewer #1: No

Reviewer #3: No

---

## [Author Response · Author response to Decision Letter 2]

6 May 2025

Please find attached the relevant documents for your review: Response_to_Reviewer_Comments.

---

## [Editor Report · Decision Letter 2]

13 May 2025

Estimation Method for Impact Force Source in Thick Coal Seams and Its Application in Rockburst Risk Quantification

PONE-D-25-09536R2

Dear Dr. Zhu,

We’re pleased to inform you that your manuscript has been judged scientifically suitable for publication and will be formally accepted for publication once it meets all outstanding technical requirements.

Kind regards,

Kang Wang, Ph.D.

Academic Editor

PLOS ONE
---

## [Editor Report · Acceptance letter]

PONE-D-25-09536R2

PLOS ONE

Dear Dr. Zhu,

I'm pleased to inform you that your manuscript has been deemed suitable for publication in PLOS ONE. Congratulations! Your manuscript is now being handed over to our production team.

Kind regards,

on behalf of

Dr. Kang Wang

Academic Editor

PLOS ONE